# Optical Mode Tuning of Monolayer Tungsten Diselenide (WSe_2_) by Integrating with One-Dimensional Photonic Crystal through Exciton–Photon Coupling

**DOI:** 10.3390/nano12030425

**Published:** 2022-01-27

**Authors:** Konthoujam James Singh, Hao-Hsuan Ciou, Ya-Hui Chang, Yen-Shou Lin, Hsiang-Ting Lin, Po-Cheng Tsai, Shih-Yen Lin, Min-Hsiung Shih, Hao-Chung Kuo

**Affiliations:** 1Department of Photonics, Institute of Electro-Optical Engineering, College of Electrical and Computer Engineering, National Yang Ming Chiao Tung University, Hsinchu 30010, Taiwan; jamesk231996@gmail.com (K.J.S.); tony51415@gmail.com (H.-H.C.); dnn1227@gmail.com (Y.-H.C.); zxc35789512@gmail.com (Y.-S.L.); 2Research Center for Applied Sciences (RCAS), Academia Sinica, Taipei 11529, Taiwan; linst8168@gmail.com (H.-T.L.); d07943004@ntu.edu.tw (P.-C.T.); shiyen@gate.sinica.edu.tw (S.-Y.L.); 3Department of Photonics, National Sun Yat-sen University, Kaohsiung 80424, Taiwan

**Keywords:** transition metal dichalogenides, excitons, light–matter interactions

## Abstract

Two-dimensional materials, such as transition metal dichalogenides (TMDs), are emerging materials for optoelectronic applications due to their exceptional light–matter interaction characteristics. At room temperature, the coupling of excitons in monolayer TMDs with light opens up promising possibilities for realistic electronics. Controlling light–matter interactions could open up new possibilities for a variety of applications, and it could become a primary focus for mainstream nanophotonics. In this paper, we show how coupling can be achieved between excitons in the tungsten diselenide (WSe_2_) monolayer with band-edge resonance of one-dimensional (1-D) photonic crystal at room temperature. We achieved a Rabi splitting of 25.0 meV for the coupled system, indicating that the excitons in WSe_2_ and photons in 1-D photonic crystal were coupled successfully. In addition to this, controlling circularly polarized (CP) states of light is also important for the development of various applications in displays, quantum communications, polarization-tunable photon source, etc. TMDs are excellent chiroptical materials for CP photon emitters because of their intrinsic circular polarized light emissions. In this paper, we also demonstrate that integration between the TMDs and photonic crystal could help to manipulate the circular dichroism and hence the CP light emissions by enhancing the light–mater interaction. The degree of polarization of WSe_2_ was significantly enhanced through the coupling between excitons in WSe_2_ and the PhC resonant cavity mode. This coupled system could be used as a platform for manipulating polarized light states, which might be useful in optical information technology, chip-scale biosensing and various opto-valleytronic devices based on 2-D materials.

## 1. Introduction

Controlling the light–matter interaction on subwavelength scales is vital for a multitude of nanotechnology applications, such as modulators, lasers, switches, waveguides, logic elements, etc. Two-dimensional (2-D) materials, such as transition metal dichalcogenides (TMDs), have attracted substantial interest in photonics with the advancement of science and technology owing to its remarkable features, such as strong excitonic effects and valley-dependent characteristics [1,2]. It is possible to control the spin and valley in monolayer TMDs due to the strong spin–orbit coupling and breaking of inversion symmetry, which is different from their bulk counterparts [3,4]. In contrast to graphene, the strong quantum confinement in the out-of-plane direction causes the bandgap of TMDs to be strongly influenced by the number of layers; the bandgap can be tuned from indirect to direct bandgap as the number of layers decreases from a few layers to a monolayer [5]. Because of their distinctive direct band gap structure, significant exciton-binding energies of a few 100 meV, and valley-associated features, such as valleytronics, these atomically thin monolayer TMDs have gained a lot of attention. The low dimensionality features of TMDs lead to an efficient light absorption and strong light–matter interactions, which is becoming significant for fundamental quantum physics. In some scenarios, excitons in TMDs that cover visible and near-infrared wavelengths can couple with surface plasmons in metal to generate plexcitons, and when the light–matter coupling is strong enough, the Rabi oscillations and strong coupling regime would result in the formation of hybrid quasi-particles, called polaritons.

Strong coupling is made possible by the local optical density of photonic states (LDOS), which is primarily determined by the spatiotemporal confinement of electromagnetic fields, as characterized by the mode volume V and resonance Q-factor, LDOS α Q/V. Because of their direct band gap and large exciton binding energy, numerous systems integrating TMDs monolayer and plasmonic nanostructures have recently demonstrated robust plasmon–exciton coupling [6,7]. However, since many of the novelties and prospective applications of these monolayer TMDs are based on their excitonic light emission, having a controllable emission in such systems is essential for developing efficient photonic components. In addition, many optical applications require a significant amount of light absorption, and photonic crystals (PhCs) are one of the most robust platforms for boosting light absorption, modulating light emission, and improving light–matter interactions [8,9,10]. The integration of TMDs with PhC structure helps to maximize the light extraction by manipulating their excitonic emissions. PhC nanocavities have an exceptionally high Q factor of up to 10^6^ and an ultrasmall mode volume (V) on the order of a cubic wavelength, allowing them to greatly increase the intensity of incident light. In general, resonant cavity can be formed in PhC by making point defects where light can be localized and trapped in the defect. The group velocity of light tends to become zero at the band edge of the PhC, resulting in the trapping of light at the band edge. When light interacts with a shock wave traveling through a one-dimensional PhC, the frequency shifts across the bandgap, narrowing the bandwidth and slowing light by orders of magnitude. This light trapping phenomenon can be used to couple band-edge resonance with other bound-state like excitons in order to achieve strong light–mater interactions. Strong coupling has been demonstrated between excitons in TMDs and optical bound states in the continuum (BICs) supported by PhC slab with a Rabi splitting of 5.4 meV [11]. The large oscillator strength of TMDs excitons, which leads to strong exciton–photon interactions and the formation of exciton polaritons, is responsible for the strong coupling in such systems. The existence of BIC in periodic PhC is triggered by the destructive interference of contrapropagating waveguide modes associated with the periodic potential. The BIC is characterized by the high-Q factor resulting in a drastic enhancement in the light–matter interaction phenomenon and is found to be beneficial in nonlinearly tunable devices [12]. 

In addition, the linear polarization of PhC might be coupled with the circular polarization (CP) of TMDs, resulting in a shift in the valley polarization of TMDs. Valleytronics in TMDs allows for the tuning of valley degrees of freedom, which opens up a lot of possibilities for encoding and manipulating data, leading to the realization of quantum devices and quantum computation [13]. However, because of the inter-valley scattering present in TMDs, the mechanism of valley depolarization is convoluted, resulting in a significant drop in the degree of valley polarization under room temperature [14]. Circular polarized light is very important for many important applications, such as circular dichroism spectroscopy [15], magnetic imaging [16], spintronics [17], quantum computing [18,19], optical communication [20], and manipulation of quantum states [21]. Controlling CP states of the light in TMDs will result in circular dichroism (CD), which is a phenomenon that occurs when light passes through a certain medium and splits into left-hand CP (LCP) and right-hand CP (RCP) polarization states. Circular dichroism manipulation is significant in a variety of platforms, particularly in display technology, and CP radiation is becoming highly prevalent in chemistry, biology, and elementary physics apart from optics [22,23,24,25,26]. Due to the existence of CD, TMDs are particularly promising materials with controllable optical chirality for facilitating the generation of CP light. It has been challenging to manipulate the valley polarization in TMDs monolayer at room temperature due to weak light matter interaction and substantial defects. However, various approaches have been adopted to enhance the CD in these TMDs using in-plane electric field [27], out-of-plane magnetic field [2], localized magnetic field [28], plasmonic structures [29], etc. It has been reported that CD in TMDs materials, such as WSe_2_, can be tuned by using plasmonic metasurfaces owing to the presence of localized surface plasmon resonance [28]. The enhanced light–matter interaction is responsible for the improvement of CD in these integrated systems. Photonic crystals can be integrated with TMDs to maximize the light extraction from TMDs through strong light–mater interactions. As a result, the integration between the monolayer TMDs and the PhC can control the polarization states of photons emitted from the TMDs. In our work, we use 1-D PhC, and its band-edge resonance is responsible for coupling with the excitons in TMDs monolayer thereby enhancing the light–mater interaction in the integrated system. 

Tungsten diselenide (WSe_2_) and molybdenum disulfide (MoS_2_) are two TMDs materials that have been the most investigated in recent years [30,31,32,33,34]. However, MoS_2_ has the downside of being easily oxidized in air, resulting in S vacancies in the material, thereby lowering its optical quality [35]. WSe_2_, on the other hand, drew a lot of attention because to its distinctive features, such as high quantum yield, strong spin-orbit coupling, and ambipolar charge transport. In addition, the peak emission wavelength, band gap and PL intensity of WSe_2_ change with the change in the number of layers. In this paper, we show how to tune the optical mode of a 1-D PhC on a flexible PDMS substrate by increasing the PhC lattice constant. Furthermore, we use a monolayer layer of WSe_2_ with a thickness of only 0.7 nm as the gain material, which can couple with the resonant mode of the 1-D PhC cavity. The magnitude of degree of polarization is significantly enhanced through the exciton–photon interaction with the integration of 1-D PhC. This work could pave a way towards the manipulation of a degree of circular polarization in various TMDs integrated systems.

## 2. Materials and Methods

### 2.1. Fabrication of SiN_x_ 1-D Photonic Crystals

Low pressure chemical vapor deposition (LPCVD) was used to deposit SiN_x_ with a thickness of 200 nm on a silicon substrate, followed by spin coating of 300 nm thick ma-N2403 photoresist and a thin layer of espacer on our SiN_x_/Si substrate. Due to its better scale and resolution to our pattern, electron beam lithography was a good candidate for our device fabrication and was used to pattern our SiN_x_/Si substrate. For the e-beam process, the spot size and the beam voltage were set at 1.0 and 30 kV when patterning, and the exposed ma-N2403 was developed by the tetramethylammonium hydroxide (TMAH 2.38%) solution. We used high-density plasma inductively coupled plasma-reactive ion etching (ICP-RIE) dry etching system for transferring the pattern into the SiN_x_ PhC layer. After the etching was completed, the residual ma-N2403 photoresist was removed from the substrate through O_2_ plasma in ICP-RIE at 20 °C with O_2_ flow of 35 sccm for 5 min to finally obtain the SiN_x_ 1-D PhC. Figure 1a illustrates the fabrication process of the SiN_x_ PhC and its SEM image and optical microscopy image are shown in Figure 1b.

### 2.2. Transfer Process of 1-D PhC Structure on Flexible Substrate

We employed the most common PDMS elastomers, Sylgard^®^ 184 from Dow Corning^®^ (Midland, MI, USA), which has two resin components with vinyl groups (Part A) and hydrosiloxane groups (Part B), for the preparation of the flexible substrate. The PDMS elastomer was cured by mixing the two resin components A and B solutions with a volume ratio of 10:1 followed by the elimination of bubbles from PDMS in vacuum environment for 1 h. Then, we poured 2 mL of the PDMS solution into a plastic Petri dish and heated it at 75 °C for 12 min to keep the sample from sinking to the bottom and the substrate was finally half-cured. We then added 1 mL of uncured PMDS solution to finish the PDMS substrate, which was subsequently bonded to an upside down SiN_x_/Si structure and heated for 30 min at 75 °C. After the PDMS substrate was constructed, the Si substrate was removed using a diluted TMAH solution (2.38%), followed by a DI water rinse and hot plate drying to obtain the SiN_x_ pattern on the PDMS substrate. The process flow of bonding the SiN_x_ 1-D PhC structure on the PDMS substrate is demonstrated in Figure 2a and the optical microscopy image of the patterned PDMS substrate with SiN_x_ is shown in Figure 2b.

### 2.3. Transfer Process of WSe_2_ Monolayer

To transfer WSe_2_ to the flexible substrate, the monolayer WSe_2_ was first grown on a sapphire substrate by chemical vapor deposition (CVD). The process flow of the transfer process of WSe_2_ to the flexible substrate is shown in Figure 3. Before transferring, we spin-coated a layer of PMMA A5 on the top of WSe_2_ monolayer with the sapphire substrate at 1000 RPM for 1 min and baked at 100 °C for 30 min. The substrate was then immersed for 90 min in hot buffered oxide etch (BOE) at 100 °C to etch the sapphire and establish a narrow gap between the PMMA/WSe_2_ layer and sapphire substrate. The substrate was immersed in DI water after the etching procedure to remove BOE, and the PMMA/WSe_2_ layer was easily separated from the sapphire substrate and floated on the DI water surface. The PMMA/WSe_2_ layer was then picked up using our flexible substrate, and we ensured that the WSe_2_ flakes were overlapped with the 1-D PhC structures. The sample was then obliquely baked at 100 °C for 12 h before being immersed in acetone for 15 min to remove the PMMA coating. Finally, we baked the sample at 100 °C to dry the substrate, and an optical image of the WSe_2_ monolayer/1-D PhC structure on a flexible substrate is shown at the end of the last transfer step.

## 3. Results and Discussion

For many years, photonic crystals have attracted attention, and in a one-dimensional photonic bandgap, the light at the band edge has practically zero group velocity, allowing for lasing. A PhC structure with uniform periodicity can be used to couple the light with the gain material. In this work, we used a 1-D SiN_x_ PhC structure having different lattice constant that varies from 410 nm to 470 nm, and the filling factor in each lattice constant is changed. The band-edge resonance of 1-D PhC couples with the excitons of WSe_2_ monolayer with an emission wavelength of 750 nm and a bandgap of 1.65 eV. For the PhC, the wavelength corresponds to the parameters of the lattice constant as shown in Figure 4a and the wavelength increases with the period in the cavity of the PhC structure. The resonant cavity mode of the PhC structure couples with WSe_2_ excitons once it is transferred to the flexible substrate, as illustrated in Figure 4b, and the PhC peaks are similar in both cases. We measured a series of devices with different periods to verify the band-edge resonant modes in the 1-D PhCs, and we expected the normalized frequency (a/λ) to be the same in the same filling factor for a band-edge mode. As shown in Figure 4c, the wavelength increases linearly with the lattice constant and the normalized frequency is approximately 0.61, indicating that the mode is the same. In order to investigate the optical modes of the 1-D PhC on a flexible PDMS substrate, we calculated the corresponding band structure of the PhC by the plane-wave expansion (PWE) method for TE-like modes. Figure 4d shows the simulated band structure with a period of 470 nm and a width of 269 nm showing the group velocity of light with different in-plane wave vector k from along the *x*-axis to along the *y*-axis. In fact, the PhC band-edge mode is likely to occur around high-symmetry points, and the flat photonic band with low group velocity can enhance the light interaction. By comparing the experimental and simulated results, we can be certain that the circle in the band structure represents the operation mode of the PhC, which has a normalized frequency of 0.61. 

We tried to investigate the characteristics of the integrated system with various filling factors after ensuring that the PhC operating mode is a band-edge resonant mode. We used SEM to determine the size of the PhC structure (4–1 to 4–6) as shown in Figure 5a, with filling factors ranging from 0.46 to 0.58. Figure 5b depicts the spectrum of devices with varying filling factors, but the same lattice constant of 470 nm and height of 200 nm, implying that the wavelength shifts depending on the filling factor ratio. It can be seen that the cavity mode and gain material of WSe_2_ have a high anti-crossing dispersion relationship, which is an essential Rabi splitting phenomenon. When the emitter–photon interaction becomes larger than the dissipation rates of the system, this enables quantum coherent oscillations between the coupled systems and the quantum superposition between different quantum states. The Rabi splitting of the coupled system is around 25.0 meV, indicating that the band-edge resonance of 1-D PhC and the excitons in WSe_2_ monolayer were coupled successfully. We can notice that the coupling wavelength increases as the filling factor increases because the corresponding normalized frequency increases as shown in Figure 5c. In the 1-D PhC structure, the effective index increases as the filling factor increases, causing the normalized frequency to decrease and coupling wavelength to be red-shifted.

To manipulate the optical characteristics of the WSe_2_ integrating with the flexible SiN_x_ 1-D PhC structure, we used a homemade extending stage for stretching the flexible substrate as shown in Figure 6a. The sample is fixed on the stage by two clamps, and the sample can be stretched in the lateral direction by rotating the micrometer. By using an optical microscope and comparing the variation of length to the original length, we were able to measure the lattice extension in terms of percentage. When the PhC structure is stretched, the strain on SiN_x_ PhC structure and PDMS substrate can result in numerous changes. Young’s modulus for SiN_x_ and PDMS are 297 GPa and 870 KPa, respectively, hence the SiN_x_ PhC structure has less deformation than PDMS. In addition, the Poisson’s ratio while the strain is applied as shown in the inset of Figure 6a can be calculated as
(1)$$\vartheta =-\frac{d\varepsilon_{trans}}{d\varepsilon_{axial}}$$
where $d\varepsilon_{trans}$ is the axial strain while changing the length of x-direction, positive when stretching; and $d\varepsilon_{axial}$ is the short-axis strain while changing the length of y-direction, negative when stretching. Figure 6b shows that the Poisson’s ratio is approximately 0.55 after a series of measurements.

The pattern’s lattice constant in this experiment is 470 nm, which increases as the pattern stretches. The extension of the lattice constant along the x-direction is measured by optical microscope and the strain percentage is defined as X (%) = (X − X_0_)/X_0_ × 100%, where X_0_ and X are the length of the original pattern bonded on the PDMS substrate and after deformation, respectively. In this experiment, the strain percentage reflects the relative deformation in the size of a single pattern, which is 30 × 30 µm^2^. The period increases from 470 nm to 506 nm as the pattern’s strain increases to 8%, as seen in Table 1. In addition, as the strain of the pattern decreases to 1.6%, the period decreases from 470 nm to 462 nm. The pattern’s period is linearly proportional to the pattern’s relative deformation, according to the measurements. 

We created the structure on a flexible PDMS substrate because we wanted to expand it so that the geometry of the structure could be fine-tuned, and the wavelength can change as the period is extended in the flexible substrate. Figure 7a represents the PL spectra of the coupled system with different strains as the system is stretched in the x-direction. As we can see, the peak of the PhC structure can be manipulated as the strain is increased or decreased. From Figure 7b, we can see that, with the strain variation from −1.6% to 6.3%, the wavelength can be fine-tuned and increases linearly with the red-shifted wavelength, which is attributed to the period extension of the PhC structure on the flexible substrate. However, the PL peak wavelength of WSe_2_ does not shift with the increase in the strain as shown in Figure 7c. In addition, in the Raman spectra of Figure 7d, the peak positions of E^1^_2g_ and A_1g_ do not split and remain constant as the PDMS strain increases. The main reason is due to a strain transfer problem caused by a large variation in the Young’s modulus of the PDMS (870 KPa) substrate and the monolayer WSe_2_ (258.6 GPa), resulting in a lower strain transfer efficiency. As a result, the WSe_2_ monolayer is harder than the PDMS substrate and hence the monolayer cannot be deformed by the flexible substrate. This problem can be overcome by using a flexible substrate with a Young’s modulus comparable to that of the monolayer WSe_2_, which improves the strain transfer efficiency.

The existence of circular dichroism in TMDs enables them to control their optical chirality; there is a significant interest and potential applications for chiral 2D nanomaterials. There are many approaches to maintain the CD of PL in TMDs, such as applications of electric field and magnetic fields [2,27,36]. In this paper, we show the manipulation of CD of WSe_2_ through integration with 1-D PhC structure. Since the PhC can boost light absorption and improve the light–matter interactions by manipulating the excitonic emission, it can also be utilized to manipulate the degree of circular polarization in TMDs. In general, 1-D PhC exhibit band-edge resonance where the density of states at the resonant frequency is very high. This can be utilized to couple with excitons in the monolayer WSe_2_ for an enhanced light–mater interaction, which in turn can improve the degree of circular polarization for WSe_2_. In general, the applications of 2D materials in valleytronics have become very significant, and the circular polarization state can be applied to encode data for optical communications. WSe_2_ is an appropriate candidate for inducing optical chirality with the integration of nanostructures. The band structures of TMDs consist of two inequivalent valleys, i.e., K and –K, which lead to strong spin–orbit coupling. As a result, according to the valley-dependent optical selection rule, right circular polarized light can couple to excitonic transitions in the K valley, whereas left circular polarized light can couple to excitonic transitions in the –K valley, resulting in the single handedness of the corresponding light emission. However, as shown in Figure 8, the PL emission from the WSe_2_ after RCP light excitation contains not only RCP light from K valley, but also LCP light corresponding to the –K valley. This is attributed to the inter-valley scattering of excitons between the K and –K valley originated from optical phonons induced by defects. This is the reason why WSe_2_ had a smaller degree of polarization at room temperature. However, when WSe_2_ is integrated with a 1-D PhC structure, the optical absorption is enhanced resulting in the generation of more excitons under RCP light excitation in the K valley. These excitons lead to the improvement in the optical activities that contribute to the enhancement in the degree of valley polarization. In addition, the inter-valley scattering between the K and –K valleys is reduced due to strong light–mater interactions, thereby suppressing the excitons generation in –K valley.

The degree of valley polarization can be defined using the following formula:(2)$$P=\frac{I_{R}-I_{L}}{I_{R}+I_{L}}\times 100\%$$
where *I_R_* and *I_L_* represent the intensities of right- and left-polarized light, respectively. The degree of valley polarization depends on the excitons population with their decay rates and the inter-valley scattering between the two valleys. As a result, the degree of valley polarization can be enhanced by increasing the excitons populations or decay rates and suppressing the inter-valley scattering. The presence of 1-D PhC leads to strong light–matter interactions due to the coupling between excitons and the band-edge resonance, which in turn increases the excitons’ decay rates and reduces the inter-valley scattering, thereby improving the degree of valley polarization in the WSe_2_ monolayer/1-D PhC integrated system.

In the experiment, we characterized the circular polarization of the exciton absorption from the bare WSe_2_ and the WSe_2_/1-D PhC devices. Figure 9a show the absorption spectra of the bare WSe_2_ exciton under the LCP light pumping. The exciton absorption of the bare WSe_2_ prefers left-hand polarization due to the valleytronic effect in the WSe_2_ monolayer. Figure 9b shows the absorption spectra of the exciton from the WSe_2_/1-D PhC device under the LCP light pumping. The difference between the LCP and RCP absorption of the WSe_2_ exciton was much more enhanced after the integration of 1-D PhC. Figure 9c shows the circular polarization degree (CD) of the exciton absorption estimated with Equation (2), and a CD value of approximately −2.44% was observed for the bare WSe_2_ monolayer under LCP light pumping. The CD spectrum of the WSe_2_/1-D PhC is shown in Figure 9d, and an exciton absorption with a CD value of approximately −20.44% was achieved under LCP light pumping conditions, while the LCP light absorption was higher than that of RCP light absorption for the WSe_2_/1-D PhC structure under LCP pumping due to the introduction of the band-edge resonant mode from 1-D PhC. The difference in the absorption spectra demonstrates the photoinduced nonreciprocal dichroic behavior in the WSe_2_/1-D PhC system owing to the coupling between excitons in WSe_2_ and band-edge resonance of the 1-D PhC. The mechanism related to this nonreciprocity is associated with the difference in the density of the excitons in the two valleys of WSe_2_ when coupled with the resonances of 1-D PhC. The degree of valley polarization increased from −2.44% to −20.44% under LCP light incidence. A similar phenomenon with the inverse CDs was also observed for the bare WSe_2_ and the WSe_2_/1-D PhC devices under the RCP pumping. The significantly high degree of valley polarization values from the WSe_2_/1-D PhC integrated system is attributed to the light–matter interaction in the coupled system resulting from the coupling between excitons and the PhC resonant mode. Hence, the optical chirality of the WSe_2_ monolayer emission can be manipulated by integrating the 1-D PhC structure with the WSe_2_ monolayer.

## 4. Conclusions

In summary, we investigated the coupling between the excitonic emission of a WSe2 monolayer and the band-edge resonance of 1-D PhC on a PDMS flexible substrate. The Rabi splitting of the coupled system was around 25.0 meV, indicating that coupling exists between the 1-PhC resonance and WSe_2_ excitons in the integrated system. Furthermore, we demonstrated the manipulation of the circular polarized light emission through the light–matter interaction between the WSe_2_ excitons and the PhC band-edge resonance. In particular, we demonstrated the enhancement of the degree of circular polarization from −2.44% to −20.44% under LCP light incidence after the integration of 1-D PhC. A similar behavior with inverse CD is also observed under RCP light pumping. The stronger light–matter interaction in the coupled system is shown to control the valley-dependent excitons generation thereby manipulating the valley polarization in the WSe_2_ monolayer. A device capable of efficiently modulating circular polarized photon emission was developed by combining a novel gain material with low-dimensional materials, such as the 1-D PhC structure. These results also provide a possible platform to control the circular polarized states of the chip-scale emitters for various applications, such as optical information technology, biosensing and various opto-valleytronic devices based on 2D materials.

## Figures and Tables

**Figure 1 nanomaterials-12-00425-f001:**
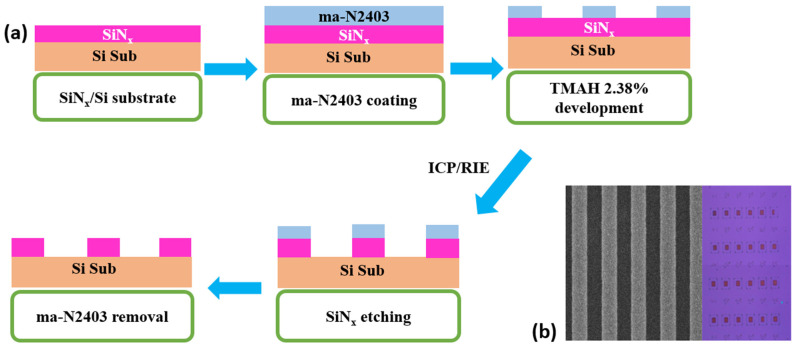
(**a**) SiN_x_ 1-D photonic crystal fabrication steps. For transferring the pattern, ICP/RIE etching technology was used. (**b**) SEM image and optical microscope image of the SiN_x_ 1-D PhC.

**Figure 2 nanomaterials-12-00425-f002:**
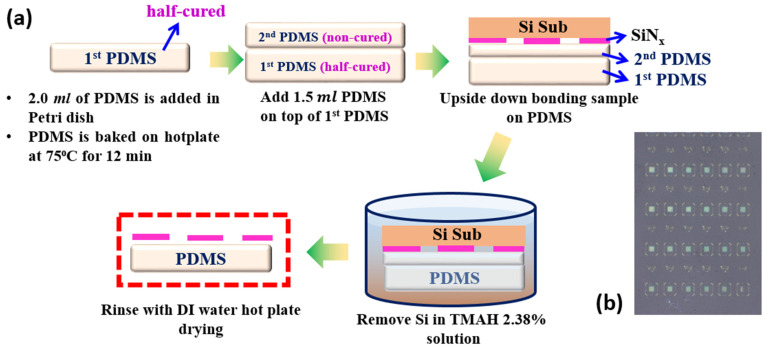
(**a**) Process flow chart of bonding the SiN_x_ pattern on PDMS. (**b**) Optical image of the fabricated structure on the PDMS substrate.

**Figure 3 nanomaterials-12-00425-f003:**
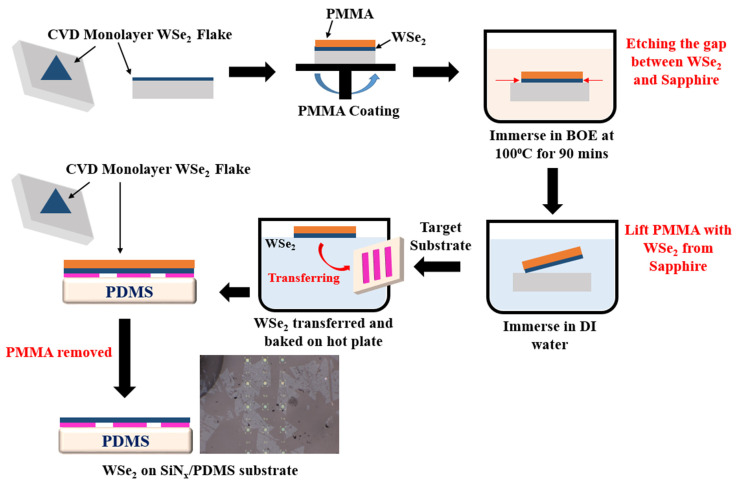
Schematic of the transfer process flow of WSe_2_ to the flexible substrate through the PMMA-assisted transfer. An optical image following the PMMA transfer for the WSe_2_ monolayer on SiN_x_/PDMS substrate is shown at the end of the last transfer step.

**Figure 4 nanomaterials-12-00425-f004:**
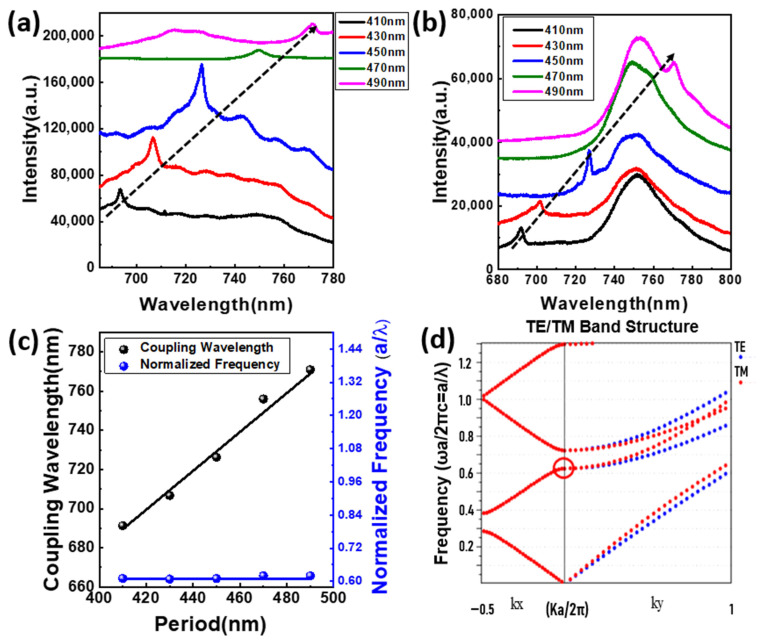
Coupling wavelength of the (**a**) 1-D PhC with different periods. (**b**) Integrated system with different lattice periods. (**c**) Coupling wavelength and normalized frequency with different lattice periods. (**d**) The 2-D PWE simulated TE-like band structure of the PhC.

**Figure 5 nanomaterials-12-00425-f005:**
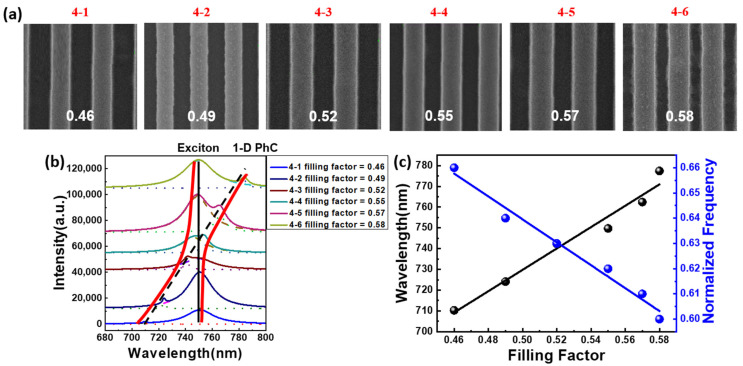
(**a**) Different 1-D PhCs (4–1 to 4–6) with various filling factors. (**b**) Spectrum of the coupled system for different filling factor ratio in the same period of 470 nm. (**c**) Coupling wavelength and corresponding normalized frequency for different filling factors.

**Figure 6 nanomaterials-12-00425-f006:**
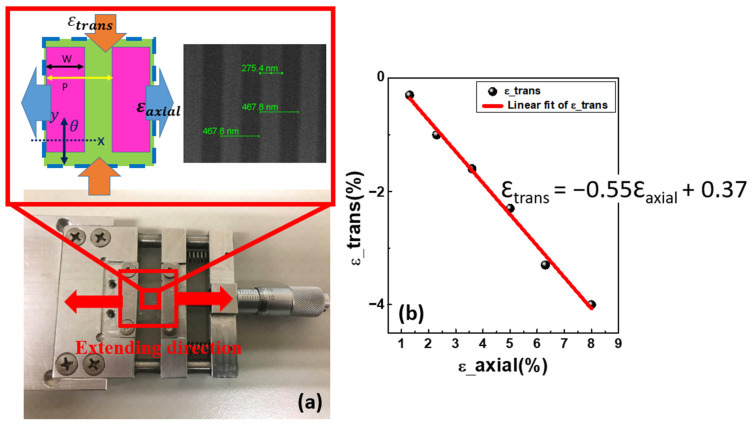
(**a**) Homemade stage that serves as the extending platform. (**b**) Poisson’s ratio calculation.

**Figure 7 nanomaterials-12-00425-f007:**
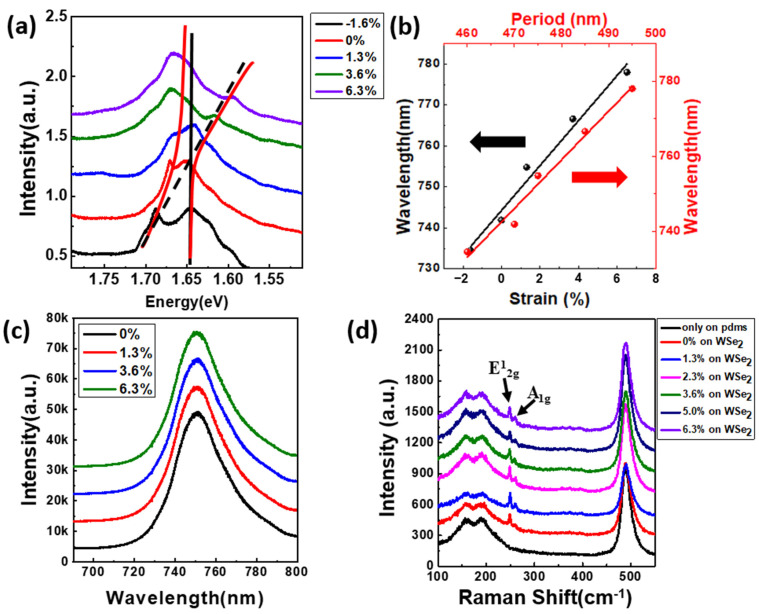
(**a**) PL spectrum of the PhC structure integrated with WSe_2_ for various strains. (**b**) Wavelength shift with strain and period. (**c**) PL spectra of WSe_2_ with different strains. (**d**) Raman spectra of WSe_2_ with different strains.

**Figure 8 nanomaterials-12-00425-f008:**
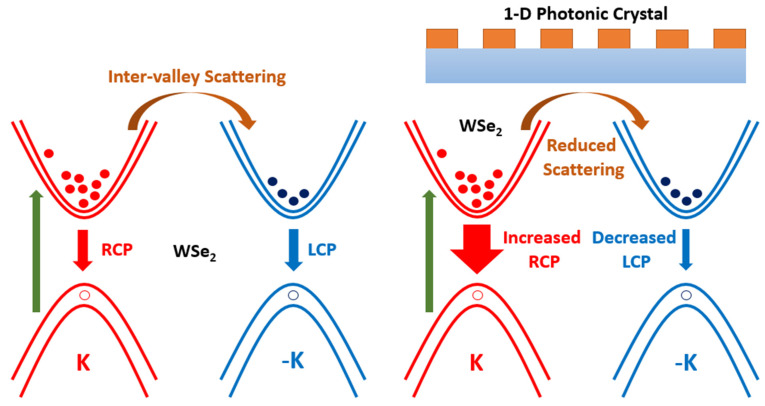
Mechanism for monitoring valley polarization in WSe_2_/1-D PhC integrated system. Under RCP light excitation, the WSe_2_/1-D PhC integrated system shows a significantly higher decay rate for excitons in K valley, while reduced decay rates for excitons in –K valley.

**Figure 9 nanomaterials-12-00425-f009:**
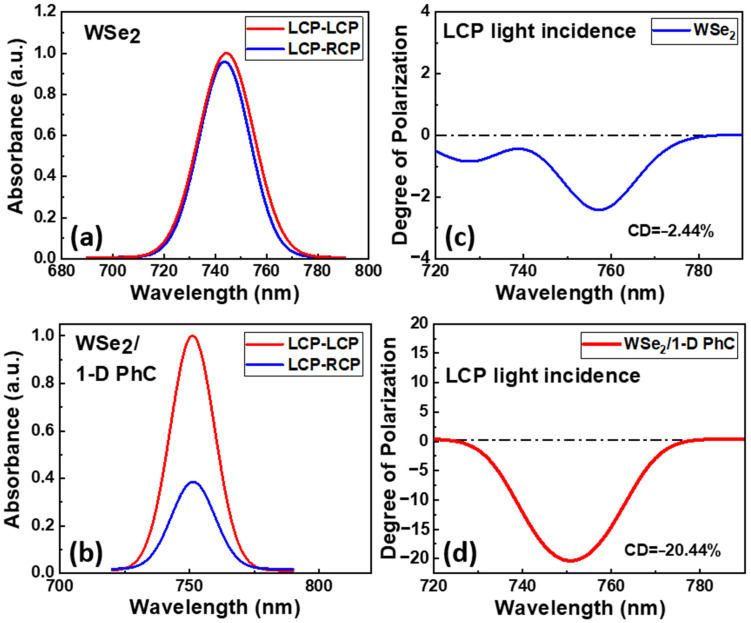
(**a**,**b**) Circularly polarized light absorption spectra for bare WSe_2_ and WSe_2_/1-D PhC under LCP light incidence. (**c**,**d**) Degree of polarization for WSe_2_ and WSe_2_/1-D PhC under LCP light incidence, respectively.

**Table 1 nanomaterials-12-00425-t001:** Estimating the period of the 1-D PhC with different strains from the optical microscope.

Strain	−1.6%	0%	1.3%	2.3%	3.6%	5.0%	6.3%	8.0%
Period	460 nm	470 nm	475 nm	479 nm	485 nm	492 nm	498 nm	506 nm

## Data Availability

Not applicable.

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
