# Peer review of "Optical Mode Tuning of Monolayer Tungsten Diselenide (WSe2) by Integrating with One-Dimensional Photonic Crystal through Exciton–Photon Coupling"

_nanomaterials, 2022, doi:10.3390/nano12030425_

Round 1

Reviewer 1 Report

The authors demonstrate the fabrication and the coupling of monolayer TMDCS with photonic nanostructures. The authors observe some really interesting optical effects due to the coupling of the two materials. The authors give an exhaustive discussion about the physical origin of this coupling.

I just have one question: in the strain effect study, did the authors consider that the strain itself induces changes to the TMDC optical response as well?

The quality of some of the figures could be improved.

Author Response

The authors demonstrate the fabrication and the coupling of monolayer TMDCS with photonic nanostructures. The authors observe some really interesting optical effects due to the coupling of the two materials. The authors give an exhaustive discussion about the physical origin of this coupling.

Response:

 We thank the reviewer for taking time to review our work and giving us fruitful suggestions and comments.

1) I just have one question: in the strain effect study, did the authors consider that the strain itself induces changes to the TMDC optical response as well?

Response:

We thank the reviewer for giving the comment and of course, reviewer is right about it and many researchers have shown the strain effect on the optical properties of thin TMDC materials as reported in https://doi.org/10.1021/nl402875m, https://doi.org/10.1103/PhysRevB.88.121301, https://doi.org/10.1038/s41377-020-00421-5, etc. However, in our study, the WSe2 monolayer is in flake form and the WSe2 flakes on the flexible PDMS substrate are also not connected with each other with non-uniform distribution. In addition, Young’s modulus for PDMS and WSe2 possess a huge difference between them leading to lower strain transfer efficiency. So, when we stretched our sample, the strain in the PDMS substrate does not have much strain effect on WSe2 monolayer flakes, as a result, there is no significant effect on optical response after the strain application. We have also given an explanation in the manuscript regarding this concerns in line 275 to 282 through Raman analysis in Figure 7(d). We also modified the sentences in the revised manuscript.

“In addition, in Raman spectra of Figure 7(d), the peak positons of E12g and A1g does not split and remains constant as the PDMS strain increases. The main reason is due to a strain transfer problem caused by a large variation in Young's modulus of the PDMS (870 KPa) substrate and the monolayer WSe2 (258.6 GPa), resulting in a lower strain transfer efficiency. As a result, the WSe2 monolayer is harder than the PDMS substrate and hence the monolayer cannot be deformed by the flexible substrate. This problem can be overcome by using a flexible substrate with Young's modulus comparable to that of the monolayer WSe2, which will improve the strain transfer efficiency.”

2)The quality of some of the figures could be improved.

Response:

We thank the reviewer for the suggestions and we have made necessary changes to improve the quality of the figures in the revised manuscript.

Reviewer 2 Report

In the work, "Optical Mode Tuning of Monolayer Tungsten Diselenide 2 (WSe2) by integrating with One-Dimensional Photonic Crystal 3 through Exciton-Photon coupling", the authors study experimentally how the coupling between excitons WSe2 monolayers with the band-edge resonance of 1D photonic crystals. They show that coupling between excitons (WSe2) and photons (photonic crystals) can be successfully coupled at room temperature and they achieve polarization control of WSe2 via this coupling.

The work is interesting and investigates properties that may be useful for quantum and optical technologies. I believe that this is a worth publishing article and I can recommend it for Nanomaterials.

Below, you will find my comments, remarks, and suggestions:

Materials and Methods

  1. Figure 1 is a bit unclear. (a) and (b) are not clearly separated and it seems as the whole second row refers to (b). A more detailed caption would be nice, as well.
  2. Line 144 mentions Fig. 1(c). Is this a typo or Fig. 1(c) is missing?
  3. For Fig. 3 see comment 1.
  4. There is no inset in Fig. 3. If you mean the last sub-picture, this is not an inset. It seems like a sequence of the process that you describe. I highly suggest restructuring this figure and providing a brief description in the caption.

Results and Discussion

  1. Lines 207-209: The authors should provide more details on which experimental and calculated results they mean. It is not clear. Also, they should elaborate more on why they are certain that the operation mode of the photonic crystal resides at the vicinity of the circle.
  2. 4. The black dashed arrows are not explained. Why they are there and what do they show? Also, axes font size should be the same in all sub-figures.
  3. Line 244 Figure (a) à 5(a)

Discussion and Conclusions

  1. Lines 360-361 do not make sense.

Conclusions should have a better description of the possible applications that their work may have. The phrase “the possibility of manipulating the circular polarized light emission, which could be used as a platform for manipulating polarized light states in optical information technology” is very vague and does not say much

Author Response

In the work, "Optical Mode Tuning of Monolayer Tungsten Diselenide 2 (WSe2) by integrating with One-Dimensional Photonic Crystal 3 through Exciton-Photon coupling", the authors study experimentally how the coupling between excitons WSe2 monolayers with the band-edge resonance of 1D photonic crystals. They show that coupling between excitons (WSe2) and photons (photonic crystals) can be successfully coupled at room temperature and they achieve polarization control of WSe2 via this coupling.

The work is interesting and investigates properties that may be useful for quantum and optical technologies. I believe that this is a worth publishing article and I can recommend it for Nanomaterials.

Response:

We thank the reviewer for taking time to review our manuscript and giving fruitful suggestions and comments.

Below, you will find my comments, remarks, and suggestions:

1)Materials and Methods

-Figure 1 is a bit unclear. (a) and (b) are not clearly separated and it seems as the whole second row refers to (b). A more detailed caption would be nice, as well.

Response:

We thank the reviewer for the suggestions and we have made necessary changes in Figure 1 and we have separated Figure 1(a) and (b). In addition, we have also made some changes in the caption to make it more clear to the readers.

-Line 144 mentions Fig. 1(c). Is this a typo or Fig. 1(c) is missing?

Response:

We thank the reviewer for sorting out the mistakes in the manuscript and we have made necessary changes in the manuscript.

-For Fig. 3 see comment 1.

Response:

We thank the reviewer for the suggestions and we have made necessary changes in Figure 3. We removed the inset section from the figure caption and we made necessary changes to be compatible with the figure.

-There is no inset in Fig. 3. If you mean the last sub-picture, this is not an inset. It seems like a sequence of the process that you describe. I highly suggest restructuring this figure and providing a brief description in the caption.

Response:

We thank the reviewer for the suggestions and we removed the inset section from the figure caption. In addition, we also made some changes in the figure caption to make it more clear.

2)Results and Discussion

Lines 207-209: The authors should provide more details on which experimental and calculated results they mean. It is not clear. Also, they should elaborate more on why they are certain that the operation mode of the photonic crystal resides at the vicinity of the circle.

Response:

We thank the reviewer for the comments. We are very sorry for this mistake; the calculated result is the simulated result in Figure 4(d) and the experimental result is the normalized frequency shown in Figure 4(c). The normalized frequency in the experimental result is calculated using the formula, f=a/λ, where a is the period of the 1-D PhC and λ is the corresponding wavelength. Hence, in both experimental and simulated results, the value of the normalized frequency is around 0.61, which is denoted by the circle in the simulated result. The normalized frequency lies at the band edge where the group velocity of light approaches zero resulting in light trapping at the band edge for better light interaction where the resonance is likely to occur. This band edge resonance is responsible for coupling with the excitons in WSe2 monolayer in the integrated system.

  1. The black dashed arrows are not explained. Why they are there and what do they show? Also, axes font size should be the same in all sub-figures.

Response:

We thank the reviewer for the comments. The black dash arrows in Figure 4(a) and (b) represents the trend of increasing wavelength for the 1-D PhC with change in cavity period. This is to differentiate the peak wavelengths change from that of WSe2 peak wavelength after the integration.

We have also made the font size of the arrows to be same in both cases.

Line 244 Figure (a) à 5(a)

Response:

We thank the reviewer for sorting out the mistake and we made necessary changes.

3)Discussion and Conclusions

Lines 360-361 do not make sense.

Response:

We thank the reviewer for the comment and we have made necessary changes. We had modified the sentences in the revised manuscript.

In summary, we investigated the coupling between the excitonic emission of a WSe2 monolayer and the band-edge resonance of 1-D PhC on a PDMS flexible substrate. The Rabi splitting of the coupled system is around 25.0 meV, indicating that coupling exist between the 1-PhC resonance and WSe2 excitons in the integrated system.

Conclusions should have a better description of the possible applications that their work may have. The phrase “the possibility of manipulating the circular polarized light emission, which could be used as a platform for manipulating polarized light states in optical information technology” is very vague and does not say much

Response:

We thank the reviewer for the suggestions and we have made necessary changes in the conclusion section. We had modified the sentences in the revised manuscript.

” These results also provide a possible platform to control circular polarized states of the chip-scale emitters for the applications such as optical information technology, bio-sensing and various opto-valleytronic devices based on 2-D materials.”

Reviewer 3 Report

The manuscript "Optical Mode Tuning of Monolayer Tungsten Diselenide (WSe2) by integrating with One-Dimensional Photonic Crystal through Exciton-Photon coupling", written by Singh et al., describes a novel approach towards an integration between the TMDs and photonic crystals and its infulence on optical properties such as circular dichroism.

The manuscript is well written, the statements are supported by results. The selected theme can be potentially important for a broad community of researchers. I have only few comments:

1.  The prepared materials should be analyzed also using AFM to demonstrate the surface homogeneity.

2. Raman spectra on the figure 7d should be accompanied also by respective Raman maps, which will show how the surface chemistry changes over the scanned area.

Author Response

The manuscript "Optical Mode Tuning of Monolayer Tungsten Diselenide (WSe2) by integrating with One-Dimensional Photonic Crystal through Exciton-Photon coupling", written by Singh et al., describes a novel approach towards an integration between the TMDs and photonic crystals and its infulence on optical properties such as circular dichroism.

Response:

We thank the reviewer for taking time to review our work and giving fruitful suggestions and comments.

The manuscript is well written, the statements are supported by results. The selected theme can be potentially important for a broad community of researchers. I have only few comments:

  1. The prepared materials should be analyzed also using AFM to demonstrate the surface homogeneity.

Response:

We thank the reviewer for the suggestion, however, in our study, the WSe2 monolayer is in flake form and the WSe2 flakes on the flexible PDMS substrate are also not connected with each other with non-uniform distribution. The flexible PDMS substrate is soft and when we stretched the substrate, the non-uniformity of WSe2 flakes will become higher and we think the AFM tip will not be able to take a clear picture of the surface homogeneity. We take the reviewer’s suggestion and we will try to do it in our future research.

  1. Raman spectra on the figure 7d should be accompanied also by respective Raman maps, which will show how the surface chemistry changes over the scanned area.

Response:

We thank the reviewer for the suggestion, however, the resolution of our Raman instrument is in micrometer range that cannot go for nanometer resolution and the period of our 1-D PhC is in nanometer range, so we cannot take a clear picture of Raman mapping to show the structural changes. The WSe2 flakes are distributed widely across the 1-D PhC structure and we hope that the Raman spectra is clearly taken from the integrated area. In addition, since we got limited time to revise the manuscript, we will not be able to arrange another measurement and we will definitely do it in our future research. Thanks again.
